# Effect of 15 BMI-Associated Polymorphisms, Reported for Europeans, across Ethnicities and Degrees of Amerindian Ancestry in Mexican Children

**DOI:** 10.3390/ijms21020374

**Published:** 2020-01-07

**Authors:** Paula Costa-Urrutia, Carolina Abud, Valentina Franco-Trecu, Valentina Colistro, Martha Eunice Rodríguez-Arellano, Rafael Alvarez-Fariña, Víctor Acuña Alonso, Bernardo Bertoni, Julio Granados

**Affiliations:** 1Laboratorio de Medicina Genómica del Hospital Regional Lic. Adolfo López Mateos, ISSSTE, Ciudad de México 01030, Mexico; marthaeunicer@yahoo.com.mx; 2Integrigen de Mexico SAPI de CV. Patriotismo 12, Ciudad de México 06100, Mexico; caroamato@gmail.com (C.A.); planentrenar@gmail.com (R.A.-F.); 3Departamento de Ecología y Evolución, Facultad de Ciencias, Universidad de la República. Iguá 4225, Montevideo 11400, Uruguay; vfranco-trecu@fcien.edu.uy; 4Departamento de Métodos Cuantitativos, Facultad de Medicina, Universidad de la República. Avda. General Flores 2125, Montevideo 11800, Uruguay; valentinacolistro@gmail.com; 5Instituto Nacional de Antropología e Historia. Periférico Sur y Zapote, Tlalpan, Ciudad de México 14030, Mexico; victor_acuna@inah.gob.mx; 6Departamento de Genetica, Facultad de Medicina, Universidad de la República. Avda. General Flores 2125, Montevideo 11800, Uruguay; bbertoni212@gmail.com; 7División de Inmunogenética, Departamento de Trasplantes, Instituto Nacional de Ciencias Médicas y Nutrición Salvador Zubirán. Avda. Vasco de Quiroga, Ciudad de México14080, Mexico; julgrate@yahoo.com

**Keywords:** genetic susceptibility, BMI, childhood obesity, Amerindian ancestry

## Abstract

In Mexico, the genetic mechanisms underlying childhood obesity are poorly known. We evaluated the effect of loci, known to be associated with childhood body mass index (BMI) in Europeans, in Mexican children from different ethnic groups. We performed linear and logistic analyses of BMI and obesity, respectively, in Mestizos and Amerindians (Seris, Yaquis and Nahuatl speakers) from Northern (*n* = 369) and Central Mexico (*n* = 8545). We used linear models to understand the effect of degree of Amerindian ancestry (AMA) and genetic risk score (GRS) on BMI z-score. Northern Mexican Mestizos showed the highest overweight-obesity prevalence (47.4%), followed by Seri (36.2%) and Central Mexican (31.5%) children. Eleven loci (*SEC16B*/rs543874, *OLFM4*/rs12429545/rs9568856, *FTO*/rs9939609, *MC4R*/rs6567160, *GNPDA2*/rs13130484, *FAIM2*/rs7132908, *FAM120AOS*/rs944990, *LMX1B*/rs3829849, *ADAM23*/rs13387838, *HOXB5*/rs9299) were associated with BMI and seven (*SEC16B*/rs543874, *OLFM4*/rs12429545/rs9568856, *FTO*/rs9939609, *MC4R*/rs6567160, *GNPDA2* rs13130484, *LMX1B*/rs3829849) were associated with obesity in Central Mexican children. One SNP was associated with obesity in Northern Mexicans and Yaquis (*SEC16B*/rs543874). We found higher BMI z-score at higher GRS (β = 0.11, *p* = 0.2 × 10^−16^) and at lower AMA (β = −0.05, *p* = 6.8 × 10^−7^). The GRS interacts with AMA to increase BMI (β = 0.03, *p* = 6.08 × 10^−3^). High genetic BMI susceptibility increase the risk of higher BMI, including in Amerindian children.

## 1. Introduction

Childhood obesity is associated with severe health problems and premature death [1]. Mexico ranks as one of the top countries worldwide in childhood obesity, with a mean national prevalence of 34.9% for schoolchildren [2].

The importance of understanding the role of genetic and environmental factors in the variation of body composition and obesity prevalence among ethnic groups has been highlighted [3]. In fact, little is known about whether the same obesity-associated loci contribute to obesity risk across a range of ancestries, or rather if there are obesity susceptibility genes unique to specific ancestries [4].

Mexico harbors high genetic diversity in admixed Mexican Mestizos, and many Amerindian groups. Mexican genetic diversity is characterized by a north-south pattern of European–Amerindian ancestry, where clear subpopulation stratification can be found among relatively isolated Amerindian groups. Meanwhile, the Mexican Mestizo subpopulations are widely distributed among the parental European/Amerindian groups. Hence, the analysis of genetic susceptibility is particularly challenging due to cultural, ethnic and genetic diversity [5,6]. From a biomedical point of view, it is important to understand whether a given set of loci show variation in susceptibility effect along the whole range of ancestry. In this sense, greater understanding of genetic obesity variation will allow deeper understanding of obesity phenotype. 

Regarding efforts to shed light on the genomic basis of Mexican childhood obesity, transferability analysis showed that the association with body mass index (BMI)/obesity found for 140 loci in European adults was also found in Mexican children for 23 of those loci [7,8,9,10,11]. These results suggest partial loci transferability from European adults to Mexican children.

The aims of the current study were to contribute to the genetic basis of BMI and obesity susceptibility in Mexican children through single nucleotide polymorphism (SNP) transferability of 15 BMI/obesity-associated loci identified in European children to Mexican ethnic groups and across degrees of Amerindian ancestry.

## 2. Results

Descriptive results of BMI, overweight and obesity prevalence by ethnic group are shown in Table 1. Northern Mexican Mestizos showed the highest overweight/obesity prevalence (47.4%), followed by Seris (36.2%), Central Mexicans (31.5%) (regular schools 35.3% and indigenous schools 27.6%) and Yaquis (24.1%). General descriptive data of participants are shown in Appendix A. 

None of the SNPs (ancestry-informative markers (AIMs) or candidates) showed departure from Hardy–Weinberg equilibrium after Bonferroni correction and linkage disequilibrium, including both SNPs of the *OLFM4* gene, which showed R^2^ = 0.23. Thus, all candidate SNPs were included in the analysis. Principal component analysis (PCA) showed that Northern Mexican Mestizos and Yaquis formed a spread cluster distributed between the European and Amerindian parental populations, while Seris formed a defined and tight cluster (Appendix A). Individual ancestry proportions showed that Seris had the highest mean of Amerindian ancestry (AMA) (mean 92%, range = 56%–99%). Although the Northern Mexican Mestizos and Yaquis distributed in a separated cluster, the mean ancestry between the two ethnic groups showed significant differences (AMA: in Northern Mexican Mestizos, mean = 43%, range = 17%–83%; in Yaquis, mean = 72%, range = 45%–96%; *p* = 0.001). The groups of Central Mexican children from Puebla were a spread cluster distributed between the European and Amerindian parental groups, with 82% mean AMA ancestry (range = 30%–99%). Children from regular schools showed a significantly lower AMA mean (80%, range = 30%–99%) than children from indigenous school (84%, range = 30%–99%, *p* = 0.0001).

Eleven out of 15 loci were significantly associated with BMI in Central Mexican children (*SEC16B* rs543874, *OLFM4* rs12429545, rs9568856, *FTO* rs9939609, *MC4R* rs6567160, *GNPDA2* rs13130484, *FAIM2* rs7132908, *FAM120AOS* rs944990, *LMX1B* rs3829849, *ADAM23* rs13387838, *HOXB5* rs9299). None of them were significantly associated with BMI in Northern Mexican children, nor in the Seri group (Table 2).

Seven SNPs were also associated with obesity in Central Mexican children (*SEC16B* rs543874, *OLFM4* rs12429545, rs9568856, *FTO* rs9939609, *MC4R* rs6567160, *GNPDA23* rs1330484, and *LMX1B* rs3829849). One SNP was associated with obesity in Northern Mexican Mestizos and Yaquis (*SEC16B*rs543874), and none of them was associated with Seris (Table 3). Allele frequency of the 15 SNPs in Mexican ethnic groups are shown in supporting information Appendix A. Power calculation by allele frequency and ethic groups is shown in Appendix A.

To construct the genetic risk score (GRS), we used the 15 candidate SNPs. The mean number of risk alleles per individual was 8 (SD = 2) and ranged from 1 to 16. The linear model showed that the simple effects of GRS and AMA are opposite; a higher GRS is associated with a higher BMI z-score (β = 0.11, SE = 0.01, *p* = 0.1 × 10^−16^), while a higher AMA (as a continuous variable) is associated with a lower BMI z-score (β = −0.05, SE = 0.01 *p* = 6.8 × 10^−7^) (Table 4, Figure 1). However, GRS interacts with AMA to increase BMI (β = 0.03, SE = 0.01, *p* = 0.006, Figure 1). Figure 1 shows that children with higher GRS increase the BMI z-score, and the slope is more marked in children with high AMA and high GRS. The model explains 1.6% of the BMI z-score variance.

## 3. Discussion

In this study, we showed different obesity prevalence among different ethnic groups in Mexican schoolchildren aged 5–13 years old. Eleven out of 15 BMI/obesity-associated SNPs (*SEC16B* rs543874, *OLFM4* rs12429545, rs9568856, *FTO* rs9939609, *MC4R* rs6567160, *GNPDA2* rs13130484, *FAIM2* rs7132908, *FAM120AOS* rs944990, *LMX1B* rs3829849, *ADAM23* rs13387838, *HOXB5* rs9299) found in European children were associated with BMI z-score also in Central Mexican children, and seven were associated with obesity. One SNP was associated with obesity in Northern Mexican Mestizos and Yaquis (*SEC16B* rs543874), and none of them were associated with Seris.

Differences in mean BMI and obesity prevalence among ethnic groups are, at least in part, due to the degree of westernized lifestyle and the genetic factor [4], which was evident for Northern and Central Mexican children. In the North part of the country, Seri, Yaqui and Northern Mexican Mestizo children (from Hermosillo, capital city of Sonora State) belong to different ethnicities, which differed in AMA (mean and ranges) and in obesity prevalence (Table 1). In Sonora, a US border state, signs of high cultural transition to westernized lifestyle and the genetic makeup (mean = 43% of AMA) lead urban Mexican Mestizo children to present the highest overweight/obesity prevalence (40%) in Mexico [12] (47.4% in our study). In an endpoint of our AMA range, we find the Seris (mean = 92% of AMA). They are the smallest population in the area; they have never exceeded one thousand inhabitants and they represent the North extreme point of genetic diversity. Even when they live in a natural environment, westernization permeates, changing traditional forms of nutrition and resulting in health problems, such as high adult diabetes prevalence [13]. The high obesity prevalence shown by boys in this study (25.9%) is a sign of the beginning of the abovementioned health problems in adults. In terms of AMA, the Yaquis (mean = 72% of AMA) are located between Sonoran urban Mexican Mestizos and Seris. Currently, 45,000 Yaquis are divided in eight villages in Northern Mexico, who live in rural and urban communities, and in Arizona, US [14,15]. This cultural flexibility and higher (than Seris) population size lead Yaquis to show greater admixture levels. However, their strong identity leads them to maintain certain traditions, especially in rural communities (such as the one in which the school studied herein is set), which may underlie the lower obesity prevalence observed in our Northern Mexican child population.

Regarding Central Mexico, the Nahuatl language is taught in the six indigenous schools in Puebla State included in this study. Nahuatl speakers from Puebla share an origin in the mountain range Sierra Madre Oriental, a fairly rugged region, which causes more isolation among communities and, in turn, facilitates maintenance of their language and culture [16]. Although less marked, compared to the North, there are differences in AMA between children from indigenous (mean = 84% AMA) and from regular (mean = 80% AMA) schools. Children from indigenous schools showed less obesity prevalence, likely due to differences in the degree of westernized lifestyle. In line with previous studies, although all Mexican Amerindian groups exhibit some degree of European admixture, Mestizo groups, especially those from lower socioeconomic strata, differ culturally from Mexican Amerindian populations more than genetically [17,18]. 

In line with this study, among the 11 BMI/obesity-associated SNPs in European children found to be significant in Central Mexican children, other SNPs in three of those genes contributed to BMI/obesity susceptibility in Mexican children from Mexico City (*GNPDA2* rs10938397; *MC4R* rs17782313, rs17782313, rs2168708, rs28753167; *FAIM2* rs7138803 [8,10,11]). In our study, only *SEC16B* rs543874 was replicated in Central and Northern Mexican Mestizos and Yaquis and, although not significantly, it was the SNP most strongly associated in Seris (*p* = 0.09). This holds importance in order to know whether the same BMI/obesity-associated loci contribute to BMI/obesity risk across a range of ancestries.

For the rest of the SNPs, for which no significant effect was found (*p* ≥ 0.05), two issues are important to be regarded before accepting the lack of their contribution to BMI phenotype. Firstly, linkage disequilibrium between index SNPs and causal loci in our population could be weaker than in a population with European ancestry, which may lead to a weaker association and thus not be detected. Secondly, if the size effect is very low in our population, it could only be detected by increasing the statistical power with a larger sample size [4], mainly in Northern Mexico where our sample size is low.

The BMI was influenced by GRS, AMA and the interaction of them (Figure 1). A higher GRS resulted in a higher BMI z-score, while a higher AMA led to a lower BMI z-score, but the GRS*AMA interaction increased the BMI z-score. This suggests that high BMI genetic susceptibility impacts on BMI, even in children with high AMA.

Several efforts have been made to examine the relative effects of Amerindian and European genetic admixture on obesity [19,20,21]. However, direct comparisons among them are difficult due to differences in the range of European–Amerindian ancestry, in the definition of ethnic groups, or because similar genetic background groups living in different environmental and cultural contexts are used. Genetic admixture studies have been valuable in identifying differences in ethnicities that cannot be explained by environmental factors alone. Individuals with mixed ancestry (Asian/Europeans, Hawaiian/Europeans, Hawaiian/Asians, Latin/Europeans and Hawaiian/Asian/Europeans) have shown higher BMI than the average for their parental ethnic groups, which suggests that differences in ancestral background may partially explain ethnic differences in the prevalence of obesity [3]. Our results go in line with these observations: indigenous children from Northern and Central Mexico showed less obesity prevalence than Mestizos from the same regions. As discussed above, variation in the degree of Mestizo/Amerindian ancestry may imply lifestyle variation. Our results suggest that children showing a higher Amerindian ancestry could keep certain cultural traditions that may serve as a protection against obesity, as compared to Mestizos, except for children with high genetic BMI susceptibility.

This study has limitations that are worthy to be mentioned. The number of Northern Mexican Mestizo, Seri and Yaqui children was considerably lower than Central Mexicans. Thus, the lack of significance of several of the gene variants could be because of a lack of statistical power to detect a low effect in a low sample size (Appendix A). Also, the study collected limited information on potential confounders such as lifestyle.

## 4. Materials and Methods

We studied 8914 children aged from 5 to 13 years from state schools in Mexico: 369 from 6 schools in Sonora State, Northern Mexico, in 2016; and 8545 from 46 schools in Puebla State, Central Mexico, in 2017. Children attended regular schools (*n* = 7712) and indigenous schools: two in Sonora, both rural schools of Seri (*n* = 68) and Yaqui (*n* = 123) Amerindian groups, and five in Puebla, for Nahuatl speakers (*n* = 1011) (Appendix A). Regular schools enrol the vast majority of students in Mexico. Indigenous schools are characterized by bilingualism and biculturalism, where at least one indigenous language and culture of a particular Amerindian group is taught. Weight and height were measured by trained technicians. Children were barefoot and wore light clothes. Accuracy of the stadiometer was ±0.1 cm and ±0.01 kg. BMI was calculated (kg/m^2^), as well as the BMI z-score.

All children were recruited by the *Por tu Salud* Project, with the goal of researching genetic and environmental factors of childhood obesity [22]. As inclusion criteria, we selected children of both sexes, with obesity, overweight and normal weight, who attended the schools. In addition, all children voluntarily accepted participating, and their parents authorized their participation by signing an informed consent. All schools were provided by Secretaría de Educación de Cultura del Estado Sonora (in English, the Sonora State Secretary of Education and Culture) and Secretaría de Educación Pública del Estado Puebla (in English, the Puebla State Public Secretary of Education). Fifteen BMI/obesity-associated SNPs, previously identified in genome-wide association studies from European children, were tested in Northern and Central Mexican children. For Northern Mexican children, genomic DNA was obtained from a sample of 500 uL of whole blood. We used an automated system InviGenius^®^ and DNA Mini Kit InviMag Blood (STRATEC Molecular GmbH, Berlin, Germany). From Central Mexican children, genomic DNA was obtained from swab samples using Star Lab Hamilton automated system and DNA Swab kit (STRATEC Molecular GmbH, Germany). The 15 SNPs included in the array were: *LMX1B* rs3829849, *MC4R* rs6567160, *ADAM23* rs13387838, *ELP3* rs13253111, *FAIM2* rs7132908, *GNPDA2* rs13130484, *GPR61* rs7550711, *RAB27B* rs8092503, *SEC16B* rs543874, *OLFM4* rs12429545, rs9568856 [23], *TNNI3K* rs12041852 [24], *FTO* rs9939609 [25], *FAM120AOS* rs944990 [26], *HOXB5* rs9299. We also included 40 ancestry-informative markers (AIMs) (allele frequency differences, d = 0.4) [6]. Genotyping was performed using a nanofluidic Dynamic Array mounted on chips in the Juno system from Fluidigm Corporation (South San Francisco, CA, USA) [27]. Ten percent of samples were replicated to evaluate genotyping reproducibility. Individuals with at least 99% of the genotyping rate were used in statistical analyses.

This project was approved by the Ethics Committee of Regional Hospital Lic. Adolfo López Mateos on 6 June 2016 and 26 September 2018 under the registry numbers 433.2016 and 315.2018 respectively, from Instituto de Seguridad y Servicios Sociales de los Trabajadores del Estado (in English, Institute for Social Security and Services for State Workers) Mexico, for Secretaría de Educación de Cultura del Estado Sonora and Secretaria de Educación Pública del Estado Puebla.

### 4.1. Data Analysis

Descriptive results were presented as BMI mean and standard deviation, and overweight and obesity prevalence according to World Health Organization (WHO) BMI cut-off [28] by ethnic group. Comparisons among groups were done using Mann–Whitney and Kruskall–Wallis tests for two or more samples, respectively, in XLSTAT software (Data Analysis and Statistical Solution for Microsoft Excel, Addinsoft, Paris, France 2017).

Fisher exact tests were employed to assess Hardy–Weinberg equilibrium, and linkage disequilibrium among SNPs was tested using R^2^ coefficient. These tests were conducted separately for AIMs and for candidate SNPs, in PLINK version 1.9 software (http://pngu.mgh.harvard.edu/purcell/plink, Harvard University, Cambridge, MA, USA) [29]. Population structure and individual admixture proportion analyses were conducted using Principal Component Analysis (PCA) in EIGENSOFT 7.2.1 [30] (https://reich.hms.harvard.edu, Harvard University, MA, USA) and ADMIXTURE version 1.3 softwares (http://www.genetics.ucla.edu/software, University of California Los Angeles, Los Angeles, CA, USA) [31], respectively. These analyses were carried out comparing Mexican samples and individuals from parental populations [6]. Data on European and African individuals were obtained from the 1000 Genomes Project (http://www.internationalgenome.org/), and Amerindian genotypes were available at the server of the project that collected the samples (ftp://ftp.inmegen.gob.mx/).

### 4.2. Association Analysis

Association analysis on BMI z-score and obesity was done using linear and logistic models, respectively, adjusted by age, sex Amerindian ancestry (AMA) and marginality index, assuming an additive inheritance model in PLINK version 1.9 software (http://pngu.mgh.harvard.edu/purcell/plink Cambridge, MA, USA) [29]. SNPs were considered significant at a lower *p*-value < 0.05. No Bonferroni adjustment was applied, since these SNPs have established associations with BMI/obesity in European children [10,32].

For further assessment of SNP–BMI z-score association, we firstly constructed a genetic risk score (GRS) combining the 15 candidate SNPs. We assumed that each SNP acts independently and contributes equally to the risk of obesity under an additive inheritance model [33,34]. Genotypes were coded as 0, 1 or 2 according to the number of risk alleles for each variant. Secondly, to understand the combined effect of the AMA (used as a continuous variable) and GRS on BMI z-score, we performed a linear model. The explanatory variables included in the models were age, sex, GRS and AMA. An initial model contained all single effects and all possible interactions of such explanatory variables. Model simplification was done by stepwise deletion of the least significant terms. We evaluated the relative performance of models using the Akaike information criteria (AIC) [35], and selected the model with the lowest AIC. These selected models were validated by residual analyses.

Power calculations to detect significance at 0.05 were performed using a continuous outcome and case-control design for Northern Mexican Mestizo/Yaqui, Seri and Central Mexican ethnic groups in Quanto software version 1.2.4 (http://biostats.usc.edu/Quanto.html, University of Southern California, Los Angeles, CA, USA). Calculations were carried out for genes only, under an additive inheritance model and using the allele frequency from 0.02 to 0.50. The whole sample means and SD for BMI (BMI mean = 18.2, SD = 4.3) and overweight/obesity prevalence (34%) were used for the analysis.

## 5. Conclusions

In conclusion, 11 out of 15 BMI/obesity-associated SNPs in European children contribute to BMI susceptibility in Central Mexican children. Higher BMI z-score genetic susceptibility increased the BMI z-score risk. The AMA attenuates the BMI z-score risk, except in those children with high genetic susceptibility. This suggests that both genetic and cultural differences among ethnicities, at least, are necessary to explain differences in obesity prevalence. 

## Figures and Tables

**Figure 1 ijms-21-00374-f001:**
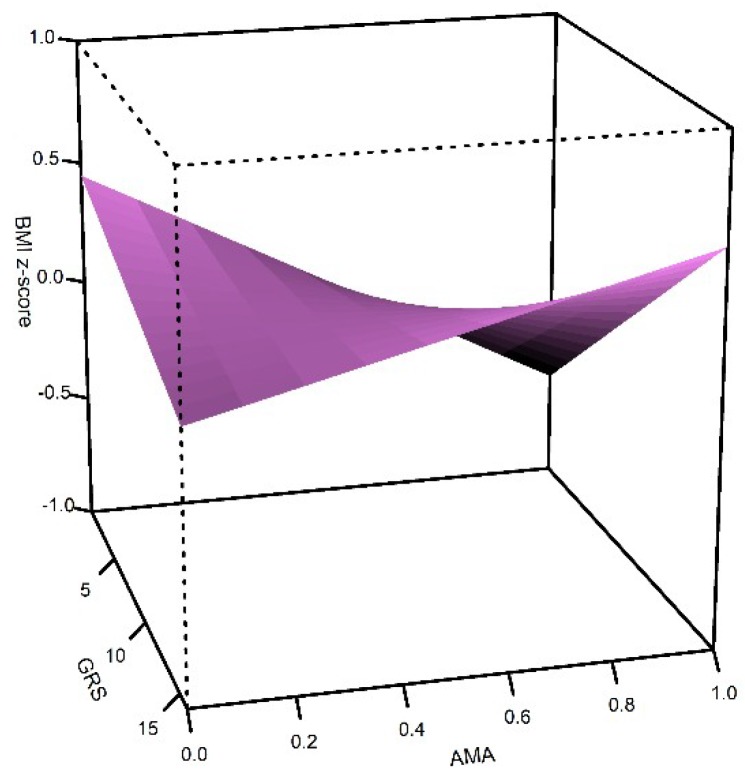
Effect of genetic risk score (GRS) and Amerindian ancestry (AMA) on body mass index z-score (BMI z-score) in Northern and Central Mexican children. Three-dimensional surfaces show the interactions among the explanatory variables in the Generalized Linear Model for BMI.

**Table 1 ijms-21-00374-t001:** Descriptive results: number of children (n), mean and standard deviation (SD) of body mass index (BMI), overweight and obesity prevalence for girls and boys of each ethnic group (NMM: Northern Mexican Mestizos, CMM: Central Mexican Mestizos).

Ethnic Group	Sex (Number)	BMI	Prevalence (%)	
		Mean (SD)	Overweight	Obesity
NMM	Girls (*n* = 100)	19.0 (4.6)	17.0	29.0
	Boys (*n* = 78)	18.9 (4.1)	20.5	28.2
Yaquis	Girls (*n* = 51)	17.3 (4.5)	9.8	7.8
	Boys (*n* = 72)	17.3 (4.2)	20.8	9.7
Seris	Girls (*n* = 41)	18.2 (3.3)	14.6	17.1
	Boys (*n* = 27)	19.2 (5.6)	14.8	25.9
CMM, indigenous school	Girls (*n* = 531)	17.2 (3.0)	11.9	15.4
	Boys (*n* = 480)	17.2 (3.1)	12.5	15.4
CMM, regular school	Girls (*n* = 3761)	17.6 (3.2)	14.6	20.3
	Boys (*n* = 3773)	17.7 (3.2)	16.4	19.3

**Table 2 ijms-21-00374-t002:** Results of association analysis with body mass index (BMI) in Northern Mexican Mestizo (NMM)/Yaqui, Seri and Central Mexican (CM) children. Significant associations are shown in bolds. Chromosome (Chr), Associated allele (AA), estimate (β), standard errors (SE), *p*-value (*p*), monomorphic (M).

Gene	Chr	SNP	AA	NMM/Yaquis (*n* = 301)		Seris (*n* = 68)	*p*	CM (*n* = 8545)	*p*
				β (SE)	*p*	β (SE)		β (SE)	
*SEC16B*	1	rs543874	G	0.21 (0.14)	0.15	−0.29 (0.25)	0.24	0.10 (0.02)	**3.4 × 10^−8^**
*OLFM4*	13	rs12429545	G	−0.20 (0.11)	0.08	0.02 (0.25)	0.94	−0.07 (0.01)	**1.2 × 10^−6^**
*FTO*	16	rs9939609	A	0.03 (0.13)	0.79	0.34 (0.36)	0.35	0.11 (0.02)	**2.4 × 10^−6^**
*MC4R*	18	rs6567160	C	−0.03 (0.20)	0.87	−0.24 (0.51)	0.64	0.13 (0.03)	**4.5 × 10^−5^**
*GNPDA2*	4	rs13130484	T	−0.08 (0.11)	0.46	0.12 (0.18)	0.53	0.06 (0.01)	**3.4 × 10^−4^**
*OLFM4*	13	rs9568856	G	0.14 (0.10)	0.19	0.05 (0.23)	0.82	−0.05 (0.01)	**1.0 × 10^−3^**
*FAIM2*	5	rs7132908	A	−0.24 (0.14)	0.08	0.12 (0.37)	0.75	0.06 (0.02)	**3.0 × 10^−3^**
*FAM120AOS*	12	rs944990	A	−0.08 (0.12)	0.48	0.32 (0.25)	0.21	0.05 (0.01)	**0.01**
*LMX1B*	9	rs3829849	A	0.05 (0.13)	0.69	0.02 (0.34)	0.99	0.06 (0.02)	**0.02**
*HOXB5*	9	rs9299	A	0.02 (0.10)	0.86	−0.36 (0.21)	0.10	0.03 (0.01)	**0.03**
*ADAM23*	17	rs13387838	G	0.01 (0.35)	0.99	M		−0.23 (0.01)	**0.04**
*ELP3*	4	rs13253111	G	−0.13 (0.10)	0.20	−0.07 (0.20)	0.73	−0.02 (0.01)	0.12
*RAB27B*	2	rs8092503	G	0.05 (0.11)	0.66	0.13 (0.28)	0.66	0.02 (0.02)	0.17
*GPR61*	4	rs7550711	T	−0.31 (0.65)	0.64	M		0.15 (0.012)	0.20
*TNNI3K*	8	rs12041852	A	0.07 (0.11)	0.53	0.67 (0.42)	0.12	−0.01 (0.01)	0.54

**Table 3 ijms-21-00374-t003:** Results of association analysis with obesity in Northern Mexican Mestizo (NMM)/Yaqui, Seri and Central Mexican (CM) children. Significant associations are shown in bold. Associated allele (AA), odds ratio (OR), confidence interval at 95% (CI), *p*-value (*p*), monomorphic (M) and very low frequency (VLF).

				NMM/Yaquis (*n* = 301)		Seris (*n* = 68)		CM (*n* = 8545)	
Gene	Chr	SNP	AA	OR (CI)	*p*	OR (CI)	*p*	OR (CI)	*p*
*SEC16B*	1	rs543874	G	**1.79 (1.04, 3.08)**	**0.04**	0.31 (0.08, 1.24)	0.09	**1.26 (1.13, 1.32)**	**1.0 × 10^−5^**
*OLFM4*	13	rs12429545	G	0.84 (0.53, 1.34)	0.47	0.44 (0.13, 1.65)	0.21	**0.85 (0.78, 0.93)**	**2.2 × 10^−4^**
*FTO*	16	rs9939609	A	0.93 (0.55, 1.57)	0.78	2.17 (0.43, 10.87)	0.34	**1.26 (1.12 1.42)**	**2.2 × 10^−4^**
*MC4R*	18	rs6567160	C	1.73 (0.79, 3.81)	0.17	1.00 (0.08, 11.99)	1.00	**1.25 (1.06, 1.48)**	**8.0 × 10^−3^**
*GNPDA2*	4	rs13130484	T	0.86 (0.54, 1.36)	0.52	1.43 (0.63, 3.25)	0.40	**1.12 (1.01, 1.21)**	**0.03**
*OLFM4*	13	rs9568856	G	1.04 (0.67, 1.59)	0.87	0.78 (0.28, 2.11)	0.62	**0.90 (0.82, 0.98)**	**0.01**
*FAIM2*	12	rs7132908	A	0.91 (0.52, 1.57)	0.73	1.41 (0.32, 6.08)	0.65	1.048 (0.93, 1.18)	0.46
*FAM120AOS*	9	rs944990	A	0.96 (0.57, 1.59)	0.78	0.42 (0.12, 1.46)	0.17	1.076 (0.97, 1.19)	0.17
*LMX1B*	9	rs3829849	A	0.92 (0.53, 1.58)	0.51	0.73 (0.15, 3.68)	0.71	**1.18 (1.02, 1.37)**	**0.03**
*ADAM23*	2	rs13387838	A	1.52 (0.44, 5.27)	0.78	M		1.45 (0.81, 2.58)	0.21
*HOXB5*	17	rs9299	G	0.94 (0.60, 1.46)	0.77	0.54 (0.20, 1.43)	0.21	0.94 (0.86, 1.02)	0.14
*ELP3*	8	rs13253111	G	0.61 (0.39, 0.94)	0.33	0.86 (0.36, 2.07)	0.74	0.94 (0.86, 1.03)	0.20
*RAB27B*	18	rs8092503	G	1.24 (0.81, 1.90)	1.00	0.42 (0.08, 2.16)	0.30	1.04 (0.96, 1.14)	0.33
*GPR61*	1	rs7550711	T	VLF		M		1.03 (0.52, 2.04)	0.93
*TNNI3K*	1	rs12041852	A	1.18 (0.74, 1.88)	0.48	4.35 (0.77, 24.43)	0.09	0.97 (0.88, 1.07)	0.54

**Table 4 ijms-21-00374-t004:** Linear model results. Estimates (β) and *p*-values (*p*) for the effects of genetic risk score (GRS) and Amerindian ancestry (AMA) on body mass index z-score in Mexican schoolchildren.

Variables	β	SE	*p*
Intercept	−0.01	0.01	0.26
GRS	0.11	0.01	0.1 × 10^−16^
AMA	−0.05	0.01	6.8 × 10^−7^
GRS*AMA	0.03	0.01	6.0 × 10^−3^

The asterisk (*) denotes the interaction of variables.

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
