# Peer review of "Effect of 15 BMI-Associated Polymorphisms, Reported for Europeans, across Ethnicities and Degrees of Amerindian Ancestry in Mexican Children"

_ijms, 2020, doi:10.3390/ijms21020374_

Round 1
Reviewer 1 Report
This study tested the association between genetic polymorphisms, which were identified in European population, with obesity in Mexican school children with various ethnic background. Significant associations were observed between some of the tested SNPs and obesity in selected regions, and that Amerindian ancestry was protective of obesity. Varying sample size between regions as well as unmeasured confounding (e.g., life style factors) might have affected the associations. These findings added evidence to the genetic mechanism underlying obesity in Mexican population.
Usually the Methods section comes before Results (unless this is a special requirement of the Journal). Regarding study sampling, did you invite all students in that school to participate? Just trying to understand the representativeness of the sample. For marginality index, do all students from the same school share the same value? If so, would you consider this a hierarchical (or multilevel) data structure, which might influence your choice of statistical methods? Different DNA testing was used for Northern Mexican children vs. Central Mexican children; does this affect the results? How were the 40 ancestry-informative markers selected? How was the genotyping reproducibility? Suggest to briefly report. It might be interesting to explore whether the associations between the tested SNPs and obesity different by gender (just for your consideration). In the end of the Results, “The model explains 1.6% of the BMI z-score variance”, what did this statistic imply? A table of descriptive of the participant characteristics would be helpful (e.g., age, sex, AMA, etc.) Page 2, line 91-92: is AMA 84% meaningfully different from 80%? The study collected limited information on potential confounders such as life style (despite AMA), would you consider this a limitation? Your conclusion in the Abstract did not well correspond to the study contents, my understanding is the genetic polymorphisms are the main focus of the study (or at least one of them). There are several grammar issues, for example: Page 2, line 80: The first sentence is not complete (but rather a phrase). Page 3, line 217-218, “We selected…. included in the study”

Author Response
Dear reviewers,
We are very thankful for your useful and constructive comments. We have carefully considered and answered all of them, and made the necessary changes in this revised manuscript. Below, we provide a detailed, point-by-point response (in bolds) to each comment and in the manuscript all and new can be tracked.
Reviewer 1
This study tested the association between genetic polymorphisms, which were identified in European population, with obesity in Mexican school children with various ethnic background. Significant associations were observed between some of the tested SNPs and obesity in selected regions, and that Amerindian ancestry was protective of obesity. Varying sample size between regions as well as unmeasured confounding (e.g., life style factors) might have affected the associations. These findings added evidence to the genetic mechanism underlying obesity in Mexican population.
Usually the Methods section comes before Results (unless this is a special requirement of the Journal). Regarding study sampling, did you invite all students in that school to participate?
Just trying to understand the representativeness of the sample. For marginality index, do all students from the same school share the same value? If so, would you consider this a hierarchical (or multilevel) data structure, which might influence your choice of statistical methods?
Authors respond. Yes, all the children belong to the same marginality index. That is because the classification of the marginality index is made by geographic zone. Besides in the public school system, children are assigned to a neighborhood’s school. At first, we took into account a possible structure in the data because of the marginality index. But we found no differences in the BMI among margination index zones. Thus we discarded any hierarchical structure. To avoid confusion, taking into account your comment, we removed the marginality index analysis from the manuscript.
Different DNA testing was used for Northern Mexican children vs. Central Mexican children; does this affect the results?
Authors respond. Samples form blood or swabs do not affect the genotyping results. The DNA extraction products have the same quality form either blood or swabs. The Stratec Company and our lab have very standardized DNA extraction by the two methods. We switched from blood to swabs (or vice versa) depending on the agreement with the Government, which, in general, depends on the number of children and the budget.
How were the 40 ancestry-informative markers selected?
Authors respond. These 40 ancestry informative markers (AIMs) were included in our array design because they have been reported as the most informative unlinked markers (allele frequency differences of d0.4) to discriminate the Europeans from the Native American population out of a panel of 260 AIMs previously validated for the Mexican population (Silva-Zolezzi et al., 2009: Analysis of genomic diversity in Mexican Mestizo populations to develop genomic medicine in Mexico PNAS May 26, 2009, 106 (21) 8611-8616; doi.org/10.1073/pnas.0903045106). We also have already used these AIMs panel in Costa-Urrutia et al 2017 (Genetic Obesity Risk and Attenuation Effect of Physical Fitness in Mexican-Mestizo Population: a Case-Control Study. Ann Hum Genet. 2017 May;81(3):106-116. doi: 10.1111/ahg.12190)
How was the genotyping reproducibility?
Authors respond. We obtained 100% of the reproducibility in 10% of the replicated samples.
Suggest to briefly report. It might be interesting to explore whether the associations between the tested SNPs and obesity different by gender (just for your consideration).
Authors respond. We explored associations by gender but we didn't found any interesting to report. A significant association was kept after stratification by gender.
In the end of the Results, “The model explains 1.6% of the BMI z-score variance”, what did this statistic imply?
Authors respond: Imply that the model explains 1.6% of the variance of the data.
A table of descriptive of the participant characteristics would be helpful (e.g., age, sex, AMA, etc.)
Authors respond. The descriptive table was added to supplementary materials as Table S1.
Page 2, line 91-92: is AMA 84% meaningfully different from 80%?
Authors respond. Yes, 84% of AMA results meaningfully different from 80%. The sample size gives enough statistical power to detect small differences
The study collected limited information on potential confounders such as life style (despite AMA), would you consider this a limitation?
Authors respond. Yes, it is a limitation. We explicitly clarified this in the limitations in the discussion section.
Your conclusion in the Abstract did not well correspond to the study contents, my understanding is the genetic polymorphisms are the main focus of the study (or at least one of them).
Authors respond. We rephrase the conclusion of the abstract to focus the conclusion con the candidate polymorphisms results.
There are several grammar issues, for example: Page 2, line 80: The first sentence is not complete (but rather a phrase). Page 3, line 217-218, “We selected…. included in the study”
Authors respond. We amended the grammar issues.
Reviewer 2 Report
This study evaluated the association of childhood BMI/obesity related loci in children of Mexican ethnicity and varying amounts of Amerindian ancestry.
Why group NMM with Yaqui for association analyses? Why not include Seri since the sample is so small? Mean ancestry differed between NMM and Yaqui so I don’t know why you grouped them as you did for association analyses. The sample sizes are so small for Seri and even the combination of NMM and Yaqui. What is your power for each association analysis? I think it would be good to add in a power analysis to the results.
Please add in the effect allele frequency to tables 2 and 3 for each snp and ancestry analysis. I also think including the total sample size would be good. Might make for too many columns but maybe the max sample size across the 15 snps could be added in a footnote?
What was the variance explained (R2) of the BMI z-score with the GRS in Northern or central Mexican children, separately? Did you check the association of the GRS score with obesity? How did this look?
What was the total sample size of GRSxAMA analysis?
Author Response
Dear reviewers,
We are very thankful for your useful and constructive comments. We have carefully considered and answered all of them, and made the necessary changes in this revised manuscript. Below, we provide a detailed, point-by-point response (in bolds) to each comment and in the manuscript all and new can be tracked.
Reviewer 2
This study evaluated the association of childhood BMI/obesity related loci in children of Mexican ethnicity and varying amounts of Amerindian ancestry.
Why group NMM with Yaqui for association analyses? Why not include Seri since the sample is so small? Mean ancestry differed between NMM and Yaqui so I don’t know why you grouped them as you did for association analyses. The sample sizes are so small for Seri and even the combination of NMM and Yaqui.
Authors respond. All is a matter of (fine-scale) population structure. We joint the NMM and Yaquis because they form a mixed population, i.e. do not form different population groups. Here we have the common pattern of mixed populations. Different groups may have different ancestry average, but individual belongs to the same general population. In the case of Seris, they are an isolated population (with very low gene flow) and form separated clusters from Yaquis and form NMM. Thus we need to treat as a separate population (Figure S2).
What is your power for each association analysis? I think it would be good to add in a power analysis to the results.
Authors respond. Power analysis by allele frequency and ethnic group was added in methods, results and we explicitly mentioned in the limitations of the study in the discussion section. A table with the minimum effect size that would be detected (β and OR) according to allele frequency, ethnic group and the parameters used were added to the supplementary material in Table S3.
Please add in the effect allele frequency to tables 2 and 3 for each snp and ancestry analysis. I also think including the total sample size would be good. Might make for too many columns but maybe the max sample size across the 15 snps could be added in a footnote?
Authors respond. The minor allele frequency (MAF) of each allele by population is in Table S2. If you consider more appropriate we can move the table from supplementary material to the main text. The sample size was added in tables 2 and 3.
What was the variance explained (R2) of the BMI z-score with the GRS in Northern or central Mexican children, separately? Did you check the association of the GRS score with obesity? How did this look?
Authors respond. The variance explained is the same for Northern and for central Mexican children. Yes, we check the association of the GRS score with obesity. The general result is the same as higher GRS higher probability of obesity.
What was the total sample size of GRSxAMA analysis?
Authors respond. The total sample size; 8,914 children